# The relationship between the nurses' work environment and the quality and safe nursing care: Slovenian study using the RN4CAST questionnaire

Martina Brešan [1], Vanja Erčulj[2,3], Jaro Lajovic[3], Mirjam Ravljen[4], Walter Sermeus[5], Štefan Grosek[6,7,8]*

1 Division of Surgery, University Medical Centre Ljubljana, Ljubljana, Slovenia, 2 Faculty of Criminal Justice and Security, University of Maribor, Ljubljana, Slovenia, 3 Ro Sigma, Research & Statistics, Ljubljana, Slovenia, 4 Faculty of Health Sciences, University of Ljubljana, Ljubljana, Slovenia, 5 Leuven Institute for Healthcare Policy, KU Leuven—University of Leuven, Leuven, Belgium, 6 Neonatal Intensive Care Unit, Neonatology Section, Division of Obstetrics and Gynecology, Department of Perinatology, University Medical Centre Ljubljana, Ljubljana, Slovenia, 7 Division of Paediatrics, Department of Paediatric Intensive Therapy, University Medical Centre Ljubljana, Ljubljana, Slovenia, 8 Department of Paediatrics, Faculty of Medicine, University of Ljubljana, Ljubljana, Slovenia

* stefan.grosek@mf.uni-lj.si, stefan.grosek@kclj.si

**Data Availability Statement:** All relevant data are within the manuscript.

## Abstract

### Introduction

The safety and quality of patient care are basic guidelines in finding new and improved solutions in nursing. Important and influential factors shape the nurses' work environment in hospitals.

### Purpose

With the study, we intended to investigate whether the perception of nurses' work environment is related to the safety culture and the quality of patient care and whether it differs according to nurses' level of education.

### Methods of work

The study with a quantitative research method was conducted at the six clinical departments of the University Medical Centre, Ljubljana in 2019. We used a survey questionnaire from the European survey Nurse forecasting in Europe (RN4CAST).

### Results

270 nurses were included in the study. The response rate was 54%. The study confirmed that there is a correlation between the assessment of the nurses' work environment and the general assessment of patient safety (r = 0.36; p <0.001), the general assessment of the quality of nursing care (r = 0.32; p <0.001), the confidence in patient self-care at discharge (r = 0.29; p <0.001) and the quality of patient care in the previous year (r = 0.27; p = 0.001).

**Funding:** The authors received no specific funding for this work.

**Competing interests:** The authors have declared that no competing interests exist.

The results showed frequent verbal abuse of nurses, in 44.9% by patients and their relatives and in 35.4% by staff. Graduate nurses rated the work environment more negatively than healthcare technicians (p = 0.003).

## Discussion and conclusion

We confirmed the correlation between the assessment of nurses' work environment and patient safety and the quality of health care, and that employees' education influences the assessment and perception of the work environment.

## Introduction

In assessing the work and organisation of public health at all levels, the work environment and, in a broader sense, the culture of the work environment, which is largely composed and co-created by nurses, are extremely important. The work environment in hospitals can increase or decrease the abilities and competencies of nurses to provide quality nursing care. Research in different health systems and different cultural settings has confirmed that factors in the work environment significantly influence the quality of nursing care and consequently the treatment outcomes of patients during hospital treatment [1–7]. Researchers [5–13] have published the results of an international study on the impact of factors in the work environment on the quality of nursing care in hospitals. This study, in which Slovenia was not included, was conducted from 2009 to 2011 in twelve European countries.

In 2018, according to the National Institute of Public Health, only 34.7% of all nursing staff in the country (21644) had a bachelor's degree [14]. Analysis of data on the Slovenian categorization of the intensity of hospital nursing shows shortage of nursing staff since 2007, and the share of the deficit of Slovenian nursing staff in hospitals has been increasing over the years (in 2013—the deficit 21.78%; in 2016—the deficit 25.40%) [15]. Analysis of data of the above-mentioned categorisation of the sample in 2011 proves that the nursing providers in Slovenian hospitals are overburdened, and there is a trend towards increasing complexity of hospital nursing [16]. Dissatisfaction with work among Slovenian nurses stems mainly from the organisation of work, non-compliance with personnel standards and norms and unregulated conditions in the field of competencies between individual professional profiles [17]. The International Organization of Nurses draws attention to the fact that an unsustainable heavy workload is also associated with increased absenteeism and employee turnover, which in turn compromises the quality of patient care [18]. In Slovenia, nurses also change to less demanding healthcare jobs, to healthcare jobs abroad or even leave the profession.

According to the presented situation in nursing, we intended to find out how nurses perceive the current situation in their work environment. The main goal of the study was to determine whether the results confirm the correlation of the work environment to the culture of patient safety in the hospital, and consequently to the quality of care. Due to the different educated hospital nursing providers, we investigated whether the perception is influenced by the level of education achieved.

## Methods

The study was based on a quantitative research method using a questionnaire for nurses (requested from the author) from the international cross-sectional study Nurse forecasting in

Europe (acronym: RN4CAST). We coordinated the work methodology with the protocol of the European project [19].

## Preparation and description of the research instrument

The survey questionnaire for nurses contained 118 questions or statements, which were divided into four sections: Your workplace, Quality and safety, Your last shift at the hospital, Your data [19]. The Slovenian translation of the instrument was performed using the standard forward-backward translation [20].

The clarity of the Slovenian translation and the uniformity of the statements and questions as regards the social environment was assessed with the assistance of a group of twenty nurses employed on the hospital department, who added comments and possible ambiguities while completing the questionnaire.

## Measures and validation of the Slovenian version of the questionnaire

The construct validity of the questionnaire was assessed by an exploratory factor analysis to determine whether similar statements measured the same latent variable as proposed by several authors [21, 22]. The set of statements in the first part of the questionnaire is designed to measure elements of the work environment based on the practice environment scale of the nursing work index (PES-NWI) [8]. All items were measured on a four-point Likert agreement scale. Four factors were extracted using the principal axis factoring method and orthogonal rotation, namely: interpersonal relationships and teamwork (16.4% of variance explained), nurses' co-decision-making and development prospects (10.3% of variance explained), organisational priorities regarding the quality of patient care (9.1% of variance explained) and support of the nursing management (8.9% of variance explained) (S1 Table). The first factor, interpersonal relationships and teamwork, included seven items. An exemplary item is "Doctors respect nurses as professionals." The second factor, nurses' co-decision-making and development prospects included five items with an exemplary item being "Registered nurses participate in the hospital's internal management". The third factor, organisational priorities regarding the quality of care included five items. The item with the highest factor weight is "There is an active quality assurance program". The fourth factor, organisational priorities regarding support of the nursing management included five items. An exemplary item is "The supervising nurse manages and leads the department well". One item "There are written, up-to-date nursing care plans for all patients."had high factor weights on two factors, and seven items did not have high factor weight on any of the factors. These items were excluded from the calculation of composite score (an average of items with high weights on a single factor). Cronbach's alpha coefficient indicated good measurement reliability for all factors (range 0.74 to 0.92) (S1 Table).

In statements about patient safety culture (measured on the five-point Likert agreement scale), two latent variables (factors) were extracted using factor analysis, namely: mutual trust (23.9% of variance explained) and the importance of patient safety and feedback (19.7% of variance explained) (S2 Table). Mutual trust included four items, which were reverse coded for the analysis. An exemplary item (reversed coded for the analysis) is "When changing shifts important information about patient care is often lost". The importance of safety included two items, which were reverse coded for the analysis. An exemplary item is "The hospital management's actions show that patient safety is one of the most important tasks". One item did not have a high factor weight on any of the factors and was excluded from the calculation of the composite score (an average of items with high weights on each factor). In this case,

Cronbach's alpha coefficient also demonstrated good reliability of the measurement (0.71 and 0.72) (S2 Table).

Work environment and quality of nursing care were evaluated by a single item on a four-point scale (from poor to excellent), while patient safety was evaluated on a five-point Likert type item (from unsatisfactory to excellent). The frequency of eight adverse events (quality indicators of healthcare) was assessed on a six-point frequency scale (never to every day).

## Sampling, inclusion and exclusion criteria

University Medical Centre Ljubljana (UMC Ljubljana) is the largest Slovenian hospital with 2150 beds. Simple random sampling of six out of ten Departments of the Division of Internal Medicine and from the nine Departments of the Division of Surgery at the UMC Ljubljana was performed using the "Simple Random Sampling Applet" computer programme [23]. Randomization was done at the department level. This is one of the probability sampling methods that yields a representative sample if the response is large and the non-response is by the chance.

By following the RN4CAST methodology and work protocol, only adult medical-surgical care nursing units were included in the study. The data of different elements of nursing practice environment to patient safety and clinical outcomes are best researched and documented within these work domains [20], thus enabling a more reliable comparison. Specialized nursing units, such as the Departments of Paediatrics, Intensive Care Units, Long-Term Care Units, Emergency Departments, Department of Anaesthetics Care and Operating Theatres were not included in the selection process of our research.

## Study and data analysis

Consent to use the questionnaire for nurses from European research was obtained in August 2018 by the European coordinator of the EU-FP7 RN4CAST-project, Full Professor Walter Sermeus, provided that we conduct a study in one hospital. Therefore we decided to conduct a study among graduate nurses and healthcare technicians at the UMC Ljubljana. In early 2019, 1721 nurses were employed in hospital departments of UMC Ljubljana, of which 47.3% were nurses with a bachelor's degree [24]. Ethical approval for the pilot study was granted by the Committee for Medical Ethics of the Republic of Slovenia (Reference no. KME RS.0120-490/2018/5). Before the start of the research work, we addressed a request for a pilot study to the head nurse of UMC Ljubljana. We also previously obtained consent for research from the hospital research group in nursing and all other necessary permits (Leadership team of the Division of Internal Medicine, Leadership team of the Division of Surgery).

The study was conducted from January 16th, 2019, to March 1st, 2019. We previously notified the head nurses and heads of selected departments by e-mail of their random selection and requested their cooperation. We presented the purpose of the study and how it would be conducted to all participating nurses in the selected departments in advance and informed them of the voluntary basis of their decision and the anonymity of participation. The survey questionnaires were handed out to participants in a sealed envelope, and a new envelope with the title of the submission was enclosed for its return. Envelopes with the completed questionnaires were sent daily from the administrative offices of individual departments with the rest of the internal mail to the one selected administrative office at the Division of Surgery. The time limit for completing and returning the questionnaires was a maximum of three weeks.

## Statistical analysis

In the study, we used the original questionnaire with all-encompassing scales, although only results of the part of the questionnaire are presented in this paper. The mean number of

patients per nurse during their last shift was calculated based on self-assessment provided by nurses. Items relating to nurses' work environment, safety culture, overall assessment of nurses' work environment (a single item), overall assessment of patient safety in the department (a single item), overall quality of nursing care in the department (a single item) and adverse events were described by percentages. Composite scores per dimension of nurses' work environment and patient safety culture (as evaluated by factor analysis) were calculated and used in multiple linear regression analysis. Two multiple regression models were built to examine the relationship between nurses' work environment and patient safety culture. Four dimensions of nurses' work environment were included as independent variables and each dimension of patient safety culture (mutual trust, importance of safety culture) was included a dependent variable in the regression model. Comparison between nurses with and without bachelor's degree in the overall assessment of the work environment was done by Mann-Whitney U test. Spearman correlation coefficient was calculated between each dimension of patient safety culture or nurses' work environment and frequency of each adverse event. Statistical testing was performed at the 0.05 significance level. No correction for multiple testing was applied. Statistical analysis was performed using SPSS software, v. 26.

No a priori sample size was determined, however post-hoc sample size calculation indicates that for achieving 80% power of the multiple linear regression model with four predictors at significance level α equal to 0.05 and effect size as determined by $R^2$ equal to 0.17, the sample size of 85 would suffice.

## Results

Of the 384 nursing staff in the selected departments, 270 nurses (70%) were present at the workplace at the time of the survey. The questionnaire was sent to all present nurses in the selected departments and 147 questionnaires (54%) were completed and returned. The majority (90.5%) of the participants were female and full-time employees (94.6%). Their average (standard deviation) age was 40 (SD = 10.9) years, and 63.3% of the nurses had a bachelor's degree. A comparison of the structure of the sample, according to the level of nurses' education, shows (Table 1) that the structure of the sample statistically significantly differs from that in the population (p < 0.001). The nursers with bachelor's degrees are overrepresented in the sample. The gender structure, however, does not statistically significantly differ from that in the population (p = 0.063). In our study, the calculated ratio was 9.94 patients per nurse per work shift. We obtained this ratio based on the data from the question: "How many patients were you directly responsible for during your most recent job?" All forms of work shifts were included.

Nurses gave the highest rating to the statements that "management expects high standards of nursing care" (47.6% of respondents completely agreed and 41.5% of them agreed with the statement) and that "they work with clinically qualified nurses" (43.5% of respondents completely agreed and 44.9% of them agreed with the statement). They gave the lowest rating to the statements that "assigning patient care tasks promotes continuity of care" (24.5% of respondents completely disagreed and 29.9% of them disagreed with the statement) and that

**Table 1. Sample description.**

|  | Sample (n = 147) | Population (n = 1393) | P |
|---|---|---|---|
| **Female sex f (%)** | 133 (90.5) | 1184 (85) | 0.063 |
| **Mean (SD) age in years** | 40 (10.9) | - | |
| **Bachelor's degree** | 93 (63.3) | 600 (43.1) | < 0.001 |

"there is enough staff to get the job done" (35.4% of respondents completely disagreed and 33.3% of them disagreed with the statement) (S1 Fig).

A comparison of the percentage of nurses' responses regarding the general assessment of the work environment, patient safety and the quality of nursing care is summarized in Fig 1.

From the statements measuring the patient safety culture, nurses agreed and most with the statement "in the department, they work on preventing mistakes being repeated" (68.5% of them agreed or strongly agreed with the statement). The agreement with the reversely coded statement that "others resent their mistakes" was the lowest (34% of them agreed or strongly agreed with the statement) (Fig 2). The lowest share of strongly agree responses was regarding "the receiving of feedback on changes made in response to report events".

Among the 8 quality indicators of healthcare listed in the questionnaire as adverse events occurring to patients or employees, verbal abuse of nurses by patients and/or their relatives as well as by staff stood out. About one-fifth (19%) of the surveyed nurses perceived weekly verbal abuse by patients and/or their relatives. The share of responses confirming the occurrence of verbal abuse of nurses by patients or their relatives once to several times a month was even higher (44.9%). The occurrence of verbal abuse of nurses by staff was perceived by 12.2% of the respondents on a weekly basis, and once to several times a month by 35.4% of nurses.

The incidence of pressure sores after admission once or more per month was confirmed by nurses by 29.9%. Similarly, complaints of patients or their relatives were detected among nurses once or more per month by 29.3%. Work-related physical injuries occurred to nurses once or more per month in 20.4%. (Fig 3).

## Determining the relationship between the perception of the work environment and patient safety culture and the quality of patient care

Using multiple linear regression with the four dimensions of nurses' work environment as independent and mutual trust as one of the two dimensions of patient safety culture as dependent variable (Table 2). We found that two dimensions of nurses' work environment were

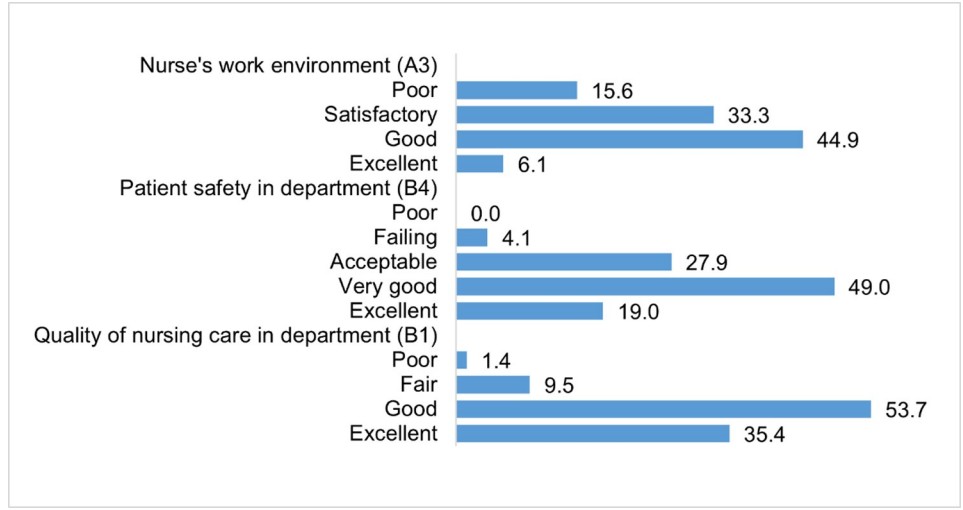

**Fig 1. Evaluation responses regarding the overall assessment of the work environment in the department, patient safety in the department and the quality of nursing care (N = 147).** A3 –How would you rate the working conditions in your current job at this hospital (e.g., suitable equipment, relations with coworkers, support from superiors); B1—How would you rate the overall quality of nursing care that patients receive in your department/unit; B4—Rate the overall safety of patients in your department/unit.

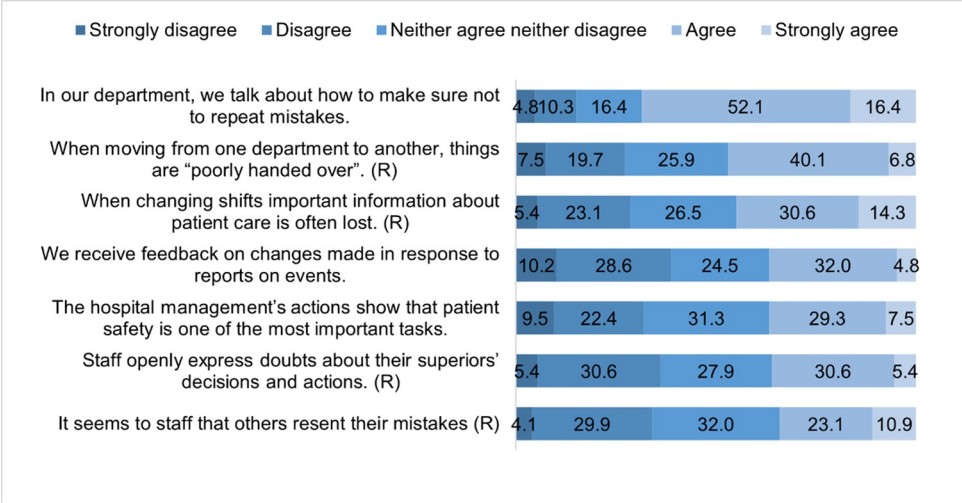

**Fig 2. Patient safety culture by statements (n = 147).**

related to mutual trust, namely organisational priorities regarding the quality of care (B = 0.30; p = 0.016) and management support of nursing care (B = 0.35; p = 0.003). Both regression coefficients are positive indicating that higher organisational priorities regarding the quality of patient care and higher management support of nursing care are reflected in higher mutual trust between healthcare staff.

The second multiple regression model included the four dimensions of nurses' work environment as independent and the second dimension of patient safety culture, namely the importance of patient safety, as a dependent variable (Table 3). Results indicate that the importance of patient safety is positively related to the nurses' co-decision-making and the opportunity for development (B = 0.31; p = 0.048) and with organisational priorities regarding the quality of patient care (B = 0.49; p = 0.001). Higher importance of patient safety is present when nurses have the opportunity for professional development and cooperate in decision

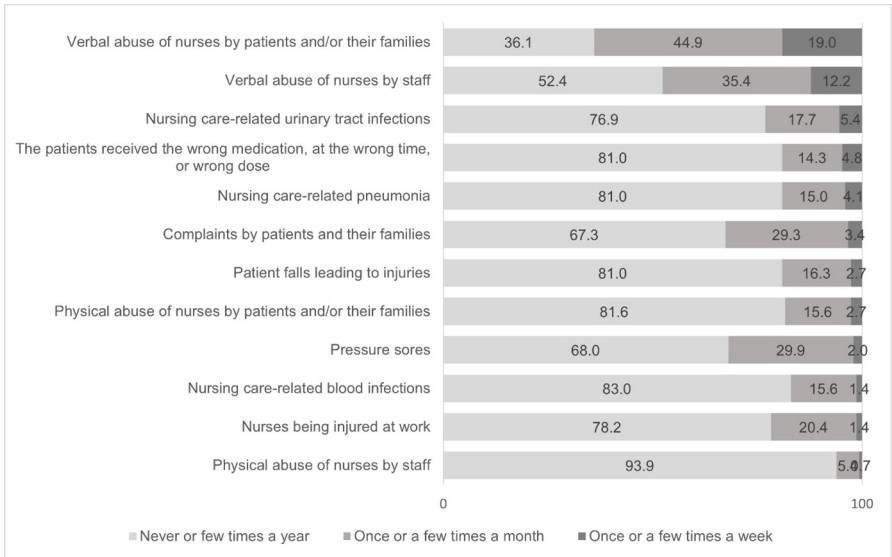

**Fig 3. Frequency of adverse events occurring to patients or staff (quality indicators of healthcare) (n = 147).**

**Table 2. Relationship between perception of the dimensions of the work environment and the mutual trust, the first dimension of the patient safety culture (result of multiple linear regression).**

|  | Regression coefficient | P |
|---|---|---|
| Intercept | 1.78 | <0.001 |
| Interpersonal relationships and teamwork | -0.06 | 0.609 |
| Nurses' co-decision-making and the opportunity for development | -0.05 | 0.693 |
| Organisational priorities regarding the quality of patient care | 0.30 | **0.016** |
| Management support nursing care | 0.35 | **0.003** |

$R^2$ = 0.17

making. The importance of patient safety goes hand in hand with firmly set priorities regarding the quality of patient care.

We found a positive correlation between all dimensions of the work environment and nurses' certainty that patients can care for themselves at discharge (S3 Table). We also confirmed a positive correlation between most dimensions of the work environment and the general assessment of the quality of nursing care in the department as well as the overall assessment of the quality of hospital care of patients in the past year (S3 Table).

Adverse events involving patients or nurses showed a negative correlation with at least two of the dimensions of the nurses' perception of the work environment (Table 4). The correlation coefficients, however, indicate a weak relationship between variables. The highest, but still moderate, negative correlation exists between verbal abuse of nurses by staff and management support of nursing care (r = 0.38; p < 0.001) and the former and overall assessment of the work environment in the hospital (r = - 0.37; p < 0.001).

## Perception of the work environment and nurses' education

We then investigated whether there is a difference in the perception of the work environment according to nurses' acquired education (with a bachelor's degree/ without a bachelor's degree). The study involved 63.3% graduate nurses (at the time of the study, their proportion at the hospital's departments was 43.1%). Nurses with a bachelor's degree gave a lower overall assessment of the work environment (U-value = 1821.5, p = 0.003). Also nurses without a bachelor's degree rated the dimension organizational priorities regarding the quality of patient care more positively than nurses with bachelor's degrees (S4 Table).

In Slovenia, nursing education takes place in two stages, in accordance with the standards of the European Directive 2013/55/EU. The professional competencies of the graduate nurse as a healthcare provider in the healthcare team comply with the competencies of the above-mentioned directive after completing 3-years of the first cycle of Bologna higher education.

**Table 3. Relationship between perception of the dimensions of the work environment and the importance of patient safety, the second dimension of the patient safety culture (results of multiple linear regression).**

|  | Regression coefficient | P |
|---|---|---|
| Intercept | 1.05 | 0.006 |
| Interpersonal relationships and teamwork | -0.01 | 0.939 |
| Nurses' co-decision-making and the opportunity for development | 0.31 | **0.048** |
| Organisational priorities regarding the quality of patient care | 0.49 | **0.001** |
| Management support nursing care | -0.02 | 0.907 |

$R^2$ = 0.20

**Table 4. Spearman's correlation coefficient between adverse events and perception of the work environment.**

| | Pressure ulcers after admission | | Patients' falls leading to injuries | | Healthcare-associated urinary tract infections | | Verbal abuse of nurses by patients and/or their families | | Verbal abuse of nurses by staff | |
|---|---|---|---|---|---|---|---|---|---|---|
| | r | P | r | P | r | P | r | P | r | P |
| Interpersonal relationships and teamwork | -0.24 | **0.003** | -0.18 | **0.031** | -0.21 | 0.009 | -0.07 | 0.423 | -0.11 | 0.192 |
| Nurses' co-decision-making and the opportunity for development | -0.19 | **0.024** | -0.24 | **0.004** | -0.17 | **0.042** | -0.25 | **0.003** | -0.30 | **<0.001** |
| Organisational priorities according to quality of care | -0.11 | 0.173 | -0.23 | **0.005** | -0.19 | **0.023** | -0.20 | **0.018** | -0.25 | **0.002** |
| Management supports nursing care | -0.04 | 0.654 | -0.07 | 0.417 | -0.15 | 0.070 | -0.25 | **0.002** | -0.38 | **<0.001** |
| Overall assessment of the work environment in the hospital | -0.14 | 0.097 | -0.16 | 0.051 | -0.21 | **0.010** | -0.33 | **<0.001** | -0.37 | **<0.001** |

r = Spearman's correlation coefficient

After completing four years of education, the healthcare technician is qualified to observe and monitor the patient's state of health, perform tasks as directed by a graduate nurse or medical doctor, report in the team, assist in patient activities of daily living, provide first aid, and care for the dying and deceased. His/her professional competencies enable him/her to carry out part of the nursing activities and diagnostic-therapeutic program independently, and part as a co-worker in a team led by a graduate nurse [25].

## Discussion

In this study, we used the RN4CAST questionnaire to determine the correlation between nurses' perception of the work environment and patient safety and the quality of nursing care. Our findings showed that two-third of nurses (68%) rated the patient safety culture as "good" or "excellent". The result is surprising, as it is 24 percentage points lower than the European average (92%) [5]. The evidence from a European survey supports the fact that patient safety is a key indicator of the quality of medical treatment and nursing care [5, 10, 12].

Various U.S. researchers have shown that a poorer safety culture of the hospital increased the 30-day mortality rate among patients with acute myocardial infarction [26] and that in patients with heart failure, the possibility of readmission within 30 days after discharge from hospital increased by 2–8% if certain nursing care activities were not performed [27]. Nursing activities left undone can pose a high risk to quality and safe patient care [28].

Among the statements about the patient safety culture, the lowest-rated statement that "employees receive feedback on changes made in response to reports on events" was alarming. To eliminate the shortcomings of existing work systems and processes, it is important that reporting and analysis of adverse events allow for timely corrective action [29] while considering interprofessional differences in perception of patient safety culture as well as certain influential factors (sufficient staff, lack of time) [30].

In our study, we demonstrated a correlation between the assessment of nurses' perception of the work environment and the overall assessment of patient safety. Almost one-third of the respondents rated patient safety as poor or satisfactory. Similarly, the results of a survey in twelve Slovenian hospitals in 2013 showed an overall low patient safety culture. The following elements of the safety culture were rated the worst: teamwork across hospital units, non-punitive response to errors, hospital management support for patient safety, and staffing [31]. Foreign researchers have reached conclusions that a better hospital work environment for nurses increases the chance of survival of patients with cardiac arrest by 16% [32]. In another study, it was shown that there is an 11% increased chance of 30-day survival in elderly patients on mechanical ventilation in better work environments, and if more graduate nurses are present [33].

The general assessment of the work environment showed that half of the nurses (49%) assessed the work environment as "poor" or "satisfactory". The result of the assessment of the work environment correlated with the percentage of nurses who would leave their current job in the next year due to dissatisfaction with the work environment (51%). Sermeus et al. [8] found that nurses in all participating countries had a negative perception of their work environment and expressed their intention to leave their current job in one year, the percentage varying from country to country (20% to 50%), with an average of 35.6% [9]. When looking for measures to retain nurses, the major Slovenian hospital identified the factors influencing nurses' leaving: inadequate pay (for less work in nursing or for easier work outside of nursing), lack of professional affiliation and the futility of performing a demanding profession in nursing care, poor interpersonal relationships and communication in health care, inadequate mentoring, desire to work in another field, insufficient staff, the rapid development of the profession and innovations, physical demands [34]. The increased turnover of nursing staff is a global and long-standing social problem, the causes of which are multifaceted, often similar and comparable regardless of the diversity of the geographical and cultural environment.

Compared to the overall assessment of patient safety, the quality of nursing care in the hospital was rated high in the study. The individual dimensions of the work environment, as well as its general assessment, showed a positive correlation with the general assessment of the quality of nursing care, with the nurses' confidence in patient self-care after discharge, and with a better assessment of the quality of patient care in the last year.

Quality indicators of healthcare as adverse events that occurred to patients or staff confirmed a negative correlation with nurses' perception of the work environment. Some associate the occurrence of adverse events with poor communication and poor cooperation in interdisciplinary teams, resulting in inappropriate handover protocols and a lack of recognition of and respect for the role of the individual [35], while at the same time attributing an important role to the leaders of nursing care teams [36]. A poor work environment as well as a high ratio of patient to nurse certainly increases the possibility of adverse events [37, 38]. The high ratio of the number of patients per nurse calculated in our study corresponds with the lowest assessment of agreement of the participating nurses that "there are not enough staff on the department to get the job done". According to European researchers [8], the number of nurses in individual countries is not a true indicator of the adequacy of the number of nursing staff. When there is a shortage of nursing staff, employers strive to make the most efficient use of the workforce, but research shows the correlation between long shifts or overtime work and nursing activities not carried out, which results in a poorer quality of care [11, 12]. The recent study in Slovenian hospitals also confirms the workload as the most stressful factor among nurses, which consequently reduces the quality of care [39]. Objectively measured personnel data based on the Slovenian categorization of the intensity of hospital nursing also prove the correlation of the long-term shortage of nursing staff with some national quality indicators of healthcare [40].

The most pronounced negative impact on the quality of nursing care in the study was the perceived verbal abuse of nurses by patients and/or their relatives on a monthly basis (44.9%), which was almost twice as high as the average in the European study (26%), and abuse by staff (35.4%), which was four times higher than the European average (8%) [13]. Unstable conditions in the work environment (insufficient number of graduate nurses, increased workloads, reduced support from management, poor team relations, etc.) have been shown to promote abuse of nurses and to have a greater impact than the patient population itself [41]. We have previously presented the findings of our study on the poor assessment of the work environment by almost half of the respondents. In 2018, a survey was conducted in three Slovenian healthcare institutions in various places on the prevalence of

violence among healthcare employees. Employees expressed the opinion that they had experienced verbal violence from patients or their relatives more than ten times in one year in 24.1% of cases, while 10% confirmed they had experienced violence from co-workers or superiors [42]. A previous Slovenian survey among nurses on workplace violence revealed that the most frequent perpetrators of verbal abuse were patients (39.3% of respondents) and co-workers (39.6% of respondents) [43].

European research has confirmed, among other things, that the level of nurses' education is important for reducing adverse events [10]. In our study, regarding the link between nurses' level of education and their perception of the work environment, we established a worse, negative general assessment of the work environment as well as a poorer assessment of dimension of the work environment on organizational priorities regarding the quality of care among graduate nurses. The percentage of graduate nurses who expressed a poor assessment of the work environment was almost one-third higher than healthcare technicians. Certainly, acquired knowledge, experience and professional competencies encourage responsible judgment, decision support, and high-level practice within the framework of nursing care regarding safe and quality patient care, as well as enabling critical evaluation of individual events and situations [7, 44]. In our study, nurses expressed a desire for greater educational opportunities as well as more prospects for advancement. On the other hand, 57% of respondents thought that there are enough graduate nurses on the department for quality patient care (answers "I partially agree" and "strongly agree"), which is contrary to the results of the European study, where, on average, almost 70% of the respondents expressed an opinion on the shortage of graduate nurses on the department [10].

## Strengths and limitations

The limitation of the research is that no correction was made for multiple testing, hence some false positive statistically significant associations might have been discovered. For more rigorous interpretation of results only those with p ≤ 0.001 could be interpreted as statistically significant.

Respondents' perceptions expressed their view on probabilities and predictions, which may differ from actual situations. Further research would be needed to clarify the more exposed results of our research.

As mentioned earlier, we conducted the research to investigate the relationship between the perception of the nurses' work environment and selected variables. The innovative European RN4CAST project is still relevant. It is important to research and discern how to create a positive work environment for nurses and other staff, that impacts sick leave and the recruitment and retention rates. Our research was carried out ten years after the European project, which may be a partial limitation. The data presented in the introduction confirm the deterioration of the situation in Slovenian nursing care. No decisive or substantive changes have been made in the past ten years, nor systematic improvements that could limit the conclusions of direct comparison of our results with the European project. However, in some cases, our results were more comparable to those of individual countries than to the overall average.

Our study is a good starting point for a wider study in all Slovenian hospitals to identify and predict the needs for nursing care providers in light of the current situation and all characteristics of nurses' work environment.

## Conclusion

The results of the study showed that there is a correlation between the assessment of nurses' work environment and patient safety culture and the quality of care in the hospital

environment. The assessment of the work environment differed between nurses with a degree in nursing and healthcare technicians, as the latter gave a more positive assessment.

The results of the study reveal a discrepancy between the overall assessment of the quality of nursing care and the assessment of patient safety. The findings show that despite the high assessment of the quality of nursing care in the department the indicators of quality of care confirmed a negative correlation with all dimensions of the work environment and the general assessment. Verbal abuse of nurses by patients and their relatives, as well as by staff, was the most notable.

## Supporting information

**S1 Fig. Average degree of agreement (M–arithmetic mean) and standard deviation (SD) for statements relating to the perception of the work environment.**
(DOCX)

**S1 Table. Factor weights (>0.40) obtained by factor analysis (principal axis factoring method and orthogonal rotation) on statements regarding work environment.**
(DOCX)

**S2 Table. Factor weights (>0.40) obtained by factor analysis (principal axis factoring method and orthogonal rotation) on statements regarding patient safety culture.**
(DOCX)

**S3 Table. Spearman's correlation coefficient between the general assessment of the quality of care, the assessment of self-care and the assessment of the quality of care in the previous year and the perception of the work environment.**
(DOCX)

**S4 Table. Perception of the work environment and nurses' education.**
(DOCX)

## Acknowledgments

We would like to thank the main coordinator of the European RN4CAST research project, Full Professor Walter Sermeus, for facilitating the use of the survey questionnaire for nurses and the implementation of the study according to the methodology of the European study at the University Medical Centre Ljubljana. Special thanks are also due to all nurses in the participating clinical departments in the study, who, despite their work commitments, demonstrated readiness and took time to complete the study questionnaires.

## Author Contributions

**Conceptualization:** Martina Brešan, Vanja Erčulj, Walter Sermeus, Štefan Grosek.

**Data curation:** Martina Brešan, Vanja Erčulj, Jaro Lajovic.

**Formal analysis:** Vanja Erčulj.

**Funding acquisition:** Martina Brešan, Štefan Grosek.

**Investigation:** Martina Brešan.

**Methodology:** Vanja Erčulj, Walter Sermeus, Štefan Grosek.

**Project administration:** Štefan Grosek.

**Resources:** Martina Brešan.

**Supervision:** Vanja Erčulj, Mirjam Ravljen, Walter Sermeus, Štefan Grosek.

**Validation:** Martina Brešan, Vanja Erčulj, Jaro Lajovic, Štefan Grosek.

**Visualization:** Martina Brešan, Vanja Erčulj.

**Writing – original draft:** Martina Brešan.

**Writing – review & editing:** Mirjam Ravljen, Walter Sermeus, Štefan Grosek.

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
