## [Decision Letter · Decision Letter 0]

16 Jun 2021

PONE-D-21-15371

THE RELATIONSHIP BETWEEN THE NURSES’ WORK ENVIRONMENT AND THE QUALITY AND SAFE NURSING CARE: SLOVENIAN STUDY USING THE RN4CAST QUESTIONNAIRE

PLOS ONE

Dear Dr. Grosek,

Thank you for submitting your manuscript to PLOS ONE. After careful consideration, we feel that it has merit but does not fully meet PLOS ONE’s publication criteria as it currently stands. Therefore, we invite you to submit a revised version of the manuscript that addresses the points raised during the review process. As you will see from the comments from both reviewers, the results of your study are of potential interest to publish. However, all sections of your manuscript need substantial revision before we can consider it for possible publication.

We look forward to receiving your revised manuscript.

Kind regards,

Barbara Schouten

Academic Editor

PLOS ONE

Journal Requirements:

Reviewers' comments:

Reviewer's Responses to Questions

**Comments to the Author**

1. Is the manuscript technically sound, and do the data support the conclusions?

Reviewer #1: Yes

Reviewer #2: Partly

2. Has the statistical analysis been performed appropriately and rigorously? 

Reviewer #1: Yes

Reviewer #2: No

3. Have the authors made all data underlying the findings in their manuscript fully available?

Reviewer #1: Yes

Reviewer #2: Yes

4. Is the manuscript presented in an intelligible fashion and written in standard English?

Reviewer #1: Yes

Reviewer #2: No

5. Review Comments to the Author

Reviewer #1: Thank you for asking me to review this paper which addresses and important area of research.

The abstract provides a balanced summary of what was done and the key findings.

The introduction could be strengthened as it does not provide a strong rationale for the study question and why it is important. The reasons for exploring the education of nurses should also be clear in the introduction. Appropriate permissions for the study appear to have been granted.

The technique for validation of the questionnaire (principal axis factor method with orthogonal rotation) should be referenced. The validation of this tool and factor weights take up a lot of space in this paper and could be presented in supplementary material instead (Tables 1 and 2). It could be clearer how the factors were used in the analysis.

The methods section should define all the key outcome variables and how these were measured. For example, it does not state that adverse events will be collected. The methods section should also describe exactly how patient safety and quality of nursing care were measured, as it is not clear for the reader. Methods should also explain how job satisfaction was measured as it is mentioned first on line 193. It is not clear how the sample size was determined.

I found the result section difficult to understand without referring to the original RN4CAST questionnaire. It needs to be clearer for the reader to understand all the variables being measured and presented. Many of the measures appear to have come from single questions on the RN4CAST (A3, B4 and B1) which could be made clearer to the reader. If possible, the authors should present effect sizes/absolute values as well as p values for correlation, as it is difficult to judge whether findings would be clinically important.

The discussion should have a section on study limitations and potential sources of bias. The discussion relates well to previous literature however the conclusion on line 243 that poor patient safety arises from a poor working environment is not substantiated from the data. The direction of effect has not been ascertained as this is a cross sectional study. The generalisability of findings could have been considered. The conclusion is succinct and based on the evidence presented.

Overall, I believe that the research has been conducted in a rigorous way using a defined population and it adds to new knowledge. However, the paper itself could be improved so that the methods and results sections are more clearly linked. The introduction and discussion sections could be strengthened as highlighted above.

Reviewer #2: This study applied the RN4CAST questionnaire to assess the perception of nurses’ work environment, safety culture, and quality of patient care in 6 clinical departments at one hospital in Slovenia. The authors also stratified results by nurses’ level of education. The data presented in this manuscript has good potential and is of interest to the discipline; however, the statistical analysis conducted by the authors needs revising. My main concern is that the authors treated ordinal categorical variables as continuous, which might be appropriate in some cases when using Linkert scales but should be checked to make sure the estimates are not considerably different. This is particularly relevant for modelling; I would suggest running an ordinal logistic regression in parallel to the multiple linear regression models in order to confirm the results presented in the manuscript. The manuscript would also benefit from revising the English and language used, as I found it difficult to follow at various points.

Introduction:

1) The authors state that they aimed to compare the results of their study with the international benchmark, derived from the study conducted by Aiken et al. However, they do not provide direct comparisons in the results section but rather contrast the two studies in the discussion section. I would therefore advise revising the aim of the study to not include this statement.

2) The number of beds, nurses and percentage of graduate nurses seem like results from the study as opposed to based on literature. If this is the case, please move these to the results section and add details on data source in the methods section. However, if based in published literature, please add a reference for these data.

3) Lines 72 to 74: Needs clarifying/rewording. My understanding is that the classification of hospital nursing intensity in hospitals across Slovenia was derived from data in 2016 that showed a deficit of nursing care providers in health care institutions. What are the implications of this?

Methods:

4) Were there any changes made to the original translation after the cognitive assessment was completed?

5) Were there any changes made to the questionnaire as described in lines 95-103 or were these changes applied on the data analysis stage? The authors mention in line 144 that they used the original questionnaire in its entirety. Please clarify to the readers.

6) Should add “data not shown” when mentioning the comparison of the structure of the sample to the population; however, might be worth adding it to supplementary as a table so that the readers can see the direct comparison. Also, please add the significance level (95% confidence intervals or p-value).

7) Exclusion criteria: why were these department excluded? Please provide justification.

Results:

8) How was the patient-per-nurse ratio calculated? What was the source for number of patients? What was defined as a work shift (e.g. 12 hours shifts, 8 hours shift, mix of both patterns)? Were these data from the same study period and averaged across all departments included? Need more details on how these were calculated.

9) The authors mention that assessment of nurses’ work environment was assessed by a 4-point agreement scale but results are presented as a 5-point scale. Could you please clarify this in the manuscript?

10) As the 5-point scale are ordered categorical data, might not be appropriate to express results as mean and standard deviation. It would be best to present the proportion of respondents that selected each category, perhaps easier to represent as a horizontal stacked bar figure. Similarly, table 3 could be reproduced as a horizontal bar figure, as this would provide more information to the readers (the proportion of each response for each question). You can then group 1-2 and 4-5 to comment on results in the text.

11) Table 3. Why was satisfactory/acceptable (score 3) grouped with scores 1 and 2? What was the rational for creating these two distinct groups (i.e. 1-3 and 4-5)? Were the differences observed statistically significant?

12) Table 5. What was the outcome variable in this regression? Mutual trust? I would suggest modelling this using an ordinal logistic regression as opposed to multiple linear regression. Also, “regression coefficient” can be used as header instead of “B”. The test statistic is unnecessary, as is the intercept.

13) Table 8. Not needed, the results of this comparison could be summarised in the text by simply adding the p-value. Would be helpful to reminder the reader the % for each education level from the study sample at this point.

What are the main differences between graduate nurses and licensed practical nurses? Which tasks are each of them responsible? These could affect their perceptions of the work environment and therefore it would be useful for the reader to know how these are classified/defined in Slovenia, as it could vary from country to country.

Discussion:

14) Not sure the authors can state that a poor work environment increases the ratio of the number of patients per nurse. It might, if morale is low enough to the point that it increases absenteeism but the opposite could also be true (i.e. high numbers of patients per nurse could lead to a poor work environment – or even constitute a poor work environment - due to increase workload on the limited staff).

15) Comparisons between the average European findings and those from this study are interesting but the authors should acknowledge the potential limitation that these surveys were conducted decades apart (2009 vs 2019) and therefore direct comparisons might not be appropriate (or might reflect changes that occurred throughout the years as opposed to actual differences between countries). Would also be helpful if the authors provided additional context to try to explain these differences – any other literature showing higher abuse by Slovenian healthcare staff?

16) Which dimensions were scored worse by graduate nurses compared to licensed practical nurses? What is the definition of these terms and how do they differ in terms of types of tasks performed and level of education?

It seems like some of the study findings discussed were not reported on the results section. Presenting stacked bar figures for each of the domains might help the reader follow the manuscript narrative and visualise all the results in a clear and concise manner.

17) I suggest that the authors add a strengths and limitation section. These could include the strength of having applied a standardised questionnaire that has been used Europe-wide. However, in the limitations section they should acknowledge that the European survey was conducted in 2009 to 2011 while this study took place in 2019, therefore contextual/external changes could have happened in the meantime, limiting conclusions from direct comparisons.

Minor comments:

Lines 82-84: This seems like the aim of the study and should be moved to the end of the introduction.

Line 87: Are these the domains referred to later in the manuscript? If so, the manuscript would benefit from standardising the nomenclature as either questionnaire sections or domains.

Lines 91-93 could be merged with the previous section, without the need of a subheading for this particular paragraph.

Line 97: spell out PES-NWI and use the abbreviation between brackets.

Table 1. Convention is to use dots for decimal places as opposed to comma. Need to explain text highlighted in red and add footnote detailing what F1-F4 stands for. These also apply to table 2.

Line 120: “This’s” should not be contracted, substitute for “This is” instead.

Line 184 to 187: Please rephrase these as I could not follow what the authors are trying to state.

Line 187 to 189: the authors joined very different concepts/responses in one sentence. I suggest splitting these into short clear sentences.

Lines 207 to 211 – Needs to be re-worded, I could not understand the comparisons that were made, and the results detailed. Would be useful to break down the paragraph into smaller sentences so that it is clear to the readers what is being compared to what, and results for each comparison.

6. PLOS authors have the option to publish the peer review history of their article (what does this mean?). If published, this will include your full peer review and any attached files.

Reviewer #1: No

Reviewer #2: No

---

## [Author Response · Author response to Decision Letter 0]

11 Sep 2021

Dear Editor-in-Chief and Editorial Board!

Thank you for the opportunity to correct the manuscript, thank you for instructions of editorial board and reviewers ’comments.

We have submitted all required items for resubmission. We edited the ORCID iD for the corresponding author. Prior to submission, the files in the PACE program were reviewed to meet the PLOS ONE technical requirements. The official translator proofread the manuscript.

We have thoroughly corrected the manuscript and we hope that this version will be suitable according to all the criteria for publication in PLOS ONE.

With kind regards

---

## [Decision Letter · Decision Letter 1]

10 Nov 2021

PONE-D-21-15371R1The relationship between the nurses' work environment and the quality and safe nursing care: Slovenian study using the RN4CAST questionnairePLOS ONE

Dear Dr. Grosek,

Thank you for submitting your manuscript to PLOS ONE. After careful consideration, we feel that it has merit but does not fully meet PLOS ONE’s publication criteria as it currently stands. Therefore, we invite you to submit a revised version of the manuscript that addresses the points raised during the review process.

We look forward to receiving your revised manuscript.

Kind regards,

Barbara Schouten

Academic Editor

PLOS ONE

Journal Requirements:

Reviewers' comments:

Reviewer's Responses to Questions

**Comments to the Author**

1. If the authors have adequately addressed your comments raised in a previous round of review and you feel that this manuscript is now acceptable for publication, you may indicate that here to bypass the “Comments to the Author” section, enter your conflict of interest statement in the “Confidential to Editor” section, and submit your "Accept" recommendation.

Reviewer #1: All comments have been addressed

Reviewer #2: (No Response)

2. Is the manuscript technically sound, and do the data support the conclusions?

Reviewer #1: Yes

Reviewer #2: Yes

3. Has the statistical analysis been performed appropriately and rigorously? 

Reviewer #1: Yes

Reviewer #2: Yes

4. Have the authors made all data underlying the findings in their manuscript fully available?

Reviewer #1: No

Reviewer #2: Yes

5. Is the manuscript presented in an intelligible fashion and written in standard English?

Reviewer #1: Yes

Reviewer #2: Yes

6. Review Comments to the Author

Reviewer #1: The authors have been extremely careful to address all the points I have raised and I believe this revised paper is very much better for your readers. The methods and discussion sections have undergone substantial revisions and I now believe this paper should be published as it adds to the evidence in this area and has been conducted in a rigorous way. Thank you for the opportunity to review it. Good luck to your authors.

The introduction could be strengthened as it does not provide a strong rationale for the study question and why it is important. THIS IS MUCH IMPROVED

The reasons for exploring the education of nurses should also be clear in the introduction. INCLUDED

The technique for validation of the questionnaire should be referenced. MUCH CLEARER NOW

The validation of this tool and factor weights take up a lot of space in this paper and could be presented in supplementary material instead (Tables 1 and 2). NOW MOVED WHICH IMPROVES THE PAPER

The methods section should define all the key outcome variables and how these were measured. For example, it does not state that adverse events will be collected. EXPLAINED IN MORE DETAIL NOW

Methods should also explain how job satisfaction was measured as it is mentioned first on line 193. INCLUDED NOW

It is not clear how the sample size was determined. NEW SECTION ADDED

I found the result section difficult to understand without referring to the original RN4CAST questionnaire. It needs to be clearer for the reader to understand all the variables being measured and presented. Many of the measures appear to have come from single questions on the RN4CAST (A3, B4 and B1) which could be made clearer to the reader. MUCH EASIER TO UNDERSTAND NOW

If possible, the authors should present effect sizes/absolute values as well as p values for correlation, as it is difficult to judge whether findings would be clinically important. EFFECT SIZES ADDED, READS SO MUCH BETTER NOW

The discussion should have a section on study limitations and potential sources of bias. STRENGTHS AND LIMITATIONS ADDED

The conclusion on line 243 that poor patient safety arises from a poor working environment is not substantiated from the data. REMOVED

The generalisability of findings could have been considered. NOW ADDED

I am not sure whether the authors have made all data underlying the findings in their manuscript fully available. They may have done this for the publishing team as a supplementary paper.

Reviewer #2: Thank you for revising the manuscript according to the suggestions and for providing a point-by-point reply to all comments. The manuscript is very much improved and reads well in my opinion. The added figures provide additional data and highlight the main findings from the study. The authors have successfully clarified all issues/points I made.

Please see below my suggestions for (very) minor revision:

I would suggest moving the last paragraph of the introduction to the methods section, as it describes the setting and the approvals obtained to conduct the study (e.g. could be moved to ‘study and data analysis’).

Similarly, some of the findings were presented in the methods section. I would suggest moving lines 144-155 to the results section.

Line 216 – Should this be “From the statements measuring the patient safety culture, nurses agreed most with the statement (…)”?

Lines 217 and 219 – When you say “68.5% agree and strongly agree responses” do you mean that 68.5% agree or strongly agree with the statement? Same for line 219. This is what I gather from figure 2, please reword if correct.

Lines 282-284 – I would suggest spelling out M and SD.

Line 312 – Should contain a comma instead of a full stop i.e. “In our study, we demonstrated (…)”.

Line 217 – Delete comma “Foreign researchers have reached conclusions,”

Lines 323-324 seem to repeat the same information reported in lines 321-322. If so, I would suggest deleting lines 321-322.

7. PLOS authors have the option to publish the peer review history of their article (what does this mean?). If published, this will include your full peer review and any attached files.

Reviewer #1: No

Reviewer #2: No

---

## [Author Response · Author response to Decision Letter 1]

30 Nov 2021

Dear Editors.

Responses to both reviewers regarding minor corrections are submitted under Response to Reviewer 1 and Response to Reviewer 2

---

## [Editor Report · Decision Letter 2]

3 Dec 2021

The relationship between the nurses' work environment and the quality and safe nursing care: Slovenian study using the RN4CAST questionnaire

PONE-D-21-15371R2

Dear Dr. Grosek,

We’re pleased to inform you that your manuscript has been judged scientifically suitable for publication and will be formally accepted for publication once it meets all outstanding technical requirements.

Kind regards,

Barbara Schouten

Academic Editor

PLOS ONE

---

## [Editor Report · Acceptance letter]

10 Dec 2021

PONE-D-21-15371R2 

The relationship between the nurses' work environment and the quality and safe nursing care: Slovenian study using the RN4CAST questionnaire 

Dear Dr. Grosek:

I'm pleased to inform you that your manuscript has been deemed suitable for publication in PLOS ONE. Congratulations! Your manuscript is now with our production department. 

Kind regards, 

on behalf of

Dr. Barbara Schouten 

Academic Editor

PLOS ONE